UPDATE

# Improvements in the sequencing and assembly of plant genomes

Priyanka Sharma[1], Othman Al-Dossary[1,2], Bader Alsubaie[1,2], Ibrahim Al-Mssallem[2], Onkar Nath[1], Neena Mitter[1], Gabriel Rodrigues Alves Margarido[1,4], Bruce Topp[1], Valentine Murigneux[3], Ardashir Kharabian Masouleh[1], Agnelo Furtado[1] and Robert J. Henry[1,5,*]

1 Queensland Alliance for Agriculture and Food Innovation, University of Queensland, Brisbane 4072, Australia
2 College of Agriculture and Food Sciences, King Faisal University, Al Hofuf, Saudi Arabia
3 Genome Innovation Hub, University of Queensland, Brisbane 4072, Australia
4 Departamento de Genética, Escola Superior de Agricultura "Luiz de Queiroz", Universidade de São Paulo, Piracicaba, São Paulo 13418-900, Brazil
5 Centre of Excellence for Plant Success in Nature and Agriculture, University of Queensland, Brisbane 4072, Australia

## ABSTRACT

**This article is an update to** https://doi.org/10.1093/gigascience/giaa146

Advances in DNA sequencing have made it easier to sequence and assemble plant genomes. Here, we extend an earlier study, and compare recent methods for long read sequencing and assembly. Updated Oxford Nanopore Technology software improved assemblies. Using more accurate sequences produced by repeated sequencing of the same molecule (Pacific Biosciences HiFi) resulted in less fragmented assembly of sequencing reads. Using data for increased genome coverage resulted in longer contigs, but reduced total assembly length and improved genome completeness. The original model species, *Macadamia jansenii*, was also compared with three other *Macadamia* species, as well as avocado (*Persea americana*) and jojoba (*Simmondsia chinensis*). In these angiosperms, increasing sequence data volumes caused a linear increase in contig size, decreased assembly length and further improved already high completeness. Differences in genome size and sequence complexity influenced the success of assembly. Advances in long read sequencing technology continue to improve plant genome sequencing and assembly. However, results were improved by greater genome coverage, with the amount needed to achieve a particular level of assembly being species dependent.

**Subjects** Genetics and Genomics, Molecular Genetics, Plant Genetics

**Submitted:** 01 February 2021

* Corresponding author. E-mail: robert.henry@uq.edu.au

Preprint submitted at https://doi.org/10.1101/2021.01.22.427724

Update to: https://doi.org/10.1093/gigascience/giaa146

## DATA DESCRIPTION

This article is an update on the previously published article; "Comparison of long-read methods for sequencing and assembly of a plant genome" [1].

Recent advances in DNA sequencing technology have facilitated the sequencing and assembly of plant genomes. There has been rapid growth in the number of reports of high-quality chromosome level assemblies [2]. A basal eudicot, *Macadamia jansenii*, was used to compare the range of long read sequencing and assembly technologies available in 2019 [1]. The Pacific Biosciences (PacBio) Sequel, Oxford Nanopore Technology (ONT) PromethION and Beijing Genomics Institute (BGI) single-tube Long Fragment Read platforms were used to analyse the same sample. Assembly tools were evaluated for these

data sets, and the contribution of short reads to improving assemblies was assessed [1]. Technology improvements had delivered continuing increases in the length and quality of sequence reads delivered by these platforms.

## Context

Since the original study, notable further advances have been made, with the use of repeated sequencing of the same molecule to greatly increase sequence accuracy for long reads. This allows the generation of long reads (10–25 kilobase pairs [kb]) with greater than 99.5% accuracy [3]. Comparison of long read technologies demonstrates the advantages and disadvantages of different platforms in relation to contiguity, accuracy of sequence and data analysis time [4]. We now update the earlier study to demonstrate the impact of these improvements on genome assemblies. Factors such as the volume of data (base pairs, bp) used in the assembly were explored for the *Macadamia* genome, related species and other diverse species with similar sized genomes.

## METHODS

### DNA extraction

All local, national and international guidelines and legislations were observed in obtaining samples for this study. *Macadamia jansenii* (NCBI:txid83725) DNA was prepared as described earlier [5]. Three other *Macadamia* species (*M. tetraphylla* [NCBI:txid512563], *M. ternifolia* [NCBI:txid4330] and *M. integrifolia* [NCBI:txid60698]) and jojoba (*Simmondsia chinensis* [NCBI:txid3999]) were also extracted using this method, with minor modifications (phenol was excluded from the extraction method) [6]. Avocado (*Persea americana* [NCBI:txid3435]) DNA was isolated by a modified cetyl-trimethyl ammonium bromide (CTAB) DNA extraction protocol [7, 8]. Leaf tissue (0.2 g) was ground and added to 15 ml of 2% CTAB buffer, pH 8.0, followed by 15 min incubation at 65 °C. After centrifugation at 10$g$ for 15 min, the supernatant was treated with RNAse A (10 ng/$\mu$l) and incubated at 37 °C for 30 min. Chloroform:isoamyl alcohol (24:1) washes were performed, followed by precipitation with isopropanol and 70% ethanol washes. The DNA was resuspended in ultrapure DNAse and RNAse-free water for sequencing.

### DNA sequencing and assembly

Long read sequencing was conducted as previously described [5]. Continuous long reads (CLR) were assembled using Falcon (RRID:SCR_016089) [1] for *M. jansenii* and Canu (RRID:SCR_015880, v 2.0) for the other genomes. HiFi gDNA libraries were prepared using sheared genomic DNA (~15–20 Kb). gDNA was sequenced on a PacBio Sequel II (software/chemistry v9.0.0) following diffusion loading. Sequence data was processed to generate circular consensus sequencing (CCS) reads using the default settings of the CCS application (RRID:SCR_017990, v4.2.0) in SMRT Link (RRID:SCR_002942, v9.0.0); minimum parameters for passes (3), accuracy (0.99), CCS read length (10) and maximum CCS read length (50,000). CCS reads were assembled using the Improved Phased Assembly (IPA) method (PacBio).

The IPA assembly method is available in protocols.io (Figure 1) [9].

The IPA assembly tool [10] uses the HiFi sequencing reads (high-quality consensus reads) and generates phased assembly. This produces a primary contig folder, including the main assembly and an associated contig, containing haplotigs and duplications. For all assemblies, 24 core processing units (CPUs) and 120 Gb of memory was employed.

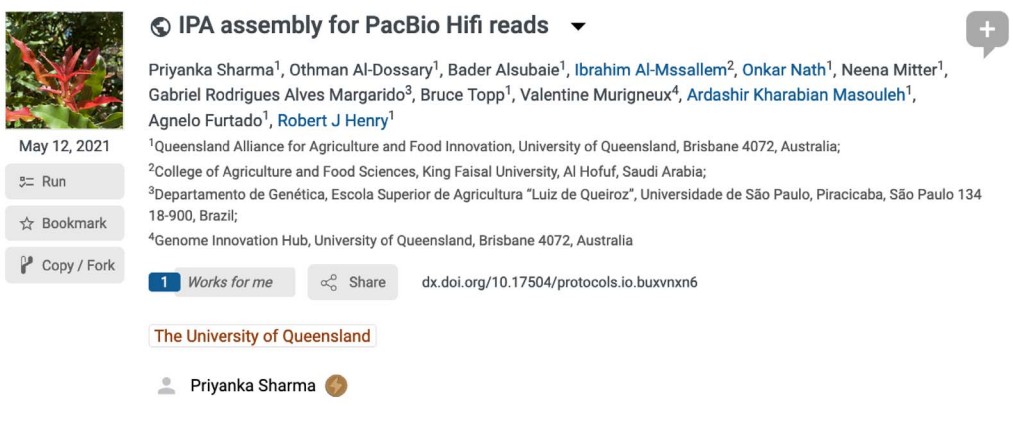

🌐 IPA assembly for PacBio Hifi reads ▾

Priyanka Sharma[1], Othman Al-Dossary[1], Bader Alsubaie[1], Ibrahim Al-Mssallem[2], Onkar Nath[1], Neena Mitter[1], Gabriel Rodrigues Alves Margarido[3], Bruce Topp[1], Valentine Murigneux[4], Ardashir Kharabian Masouleh[1], Agnelo Furtado[1], Robert J Henry[1]

[1]Queensland Alliance for Agriculture and Food Innovation, University of Queensland, Brisbane 4072, Australia;

[2]College of Agriculture and Food Sciences, King Faisal University, Al Hofuf, Saudi Arabia;

[3]Departamento de Genética, Escola Superior de Agricultura "Luiz de Queiroz", Universidade de São Paulo, Piracicaba, São Paulo 134 18-900, Brazil;

[4]Genome Innovation Hub, University of Queensland, Brisbane 4072, Australia

**Figure 1.** Protocol for IPA assembly for PacBio Hifi reads [9]. https://www.protocols.io/widgets/doi?uri=dx.doi.org/10.17504/protocols.io.buxvnxn6

**Table 1.** Improvement in long read sequencing (PacBio) for *Macadamia jansenii* when using higher accuracy sequencing.

| Parameter | Long reads* [1] | HiFi |
|---|---|---|
| Total data (Gb) | 65.2 | 28 |
| Contig N50 (Mb) | 1.57 | 4.49 |
| Assembly length (Mb) | 758 | 738 |
| Number of contigs (n) | 762 | 284 |
| BUSCO (%) | 97 | 98 |

* phased Falcon assembly. BUSCO: Benchmarking Universal Single-Copy Orthologs; Gb: gigabase pairs; Mb: megabase pairs.

## Assessment of completeness

The completeness of genome assemblies was evaluated using benchmarking universal single-copy orthologues (BUSCO) analysis (RRID:SCR_015008, v4.1.2 and v5.beta) [11, 12], using genome mode and lineage Eukaryota_odb10 dataset.

## DATA VALIDATION AND QUALITY CONTROL

## Long read versus HiFi assemblies

Comparison of assemblies based upon long reads [13] and circular consensus sequence (CCS) reads from HiFi [3] showed that greater accuracy of the CCS reads resulted in greatly improved assemblies for the *M. jansenii* genome (Table 1).

The assembly with the high quality HiFi reads was less fragmented, with slightly reduced total genome length and improved completeness (Benchmarking Universal Single-Copy Orthologs, BUSCO). The use of around 20 gigabase pairs (Gb) of high-quality (HiFi) data gave N50 values of 4 megabase pairs (Mb) and resulted in assemblies with fewer than 300 contigs required to cover the genome. This represents a considerable advance over the assemblies that were possible when this sample was previously used to compare different long read platforms and assembly tools, many of which required long computing times to assemble contigs [1]. The high-quality Improved Phased Assembler (IPA) assembly had a run time of 20 h with 120 gigabytes (GB) peak memory on the FlashLite computer cluster. This analysis requirement compares favorably with the results for a large number of earlier

**Table 2.** Comparison of assemblies of *Macadamia* species*.

| Parameter | *M. jansenii* | *M. integrifolia* | *M. tetraphylla* | *M. ternifolia* |
|---|---|---|---|---|
| Contig N50 (Mb) | 4.5 | 5.3 | 10.0 | 6.4 |
| Longest Contig (Mb) | 16.6 | 26.4 | 32.1 | 21.2 |
| Assembly Length (Mb) | 738 | 742 | 707 | 716 |
| Number of contigs (n) | 284 | 249 | 153 | 211 |
| BUSCO (%) | 97.2 | 98 | 97 | 98 |

* Primary assemblies shown. For details of associate assemblies see Table 3.

**Table 3.** Data for associated contigs in IPA assemblies.

| Parameter | *M. jansenii* | *M. integrifolia* | *M. ternifolia* | *M. tetraphylla* | Jojoba | Avocado |
|---|---|---|---|---|---|---|
| Contig N50 (Mb) | 0.45 | 1.23 | 0.77 | 1.83 | 1.69 | 1.53 |
| Longest contig (Mb) | 5.23 | 10.22 | 5.68 | 14.97 | 8.25 | 10.0 |
| Assembly length (Mb) | 527 | 671 | 590 | 655 | 738 | 788 |
| Number of contigs | 3966 | 3226 | 3006 | 2103 | 1999 | 3196 |

assembly tools reported for the same sample [1], but provides a much higher quality assembly. Assembly of the HiFi data with other recent tools was also compared. Flye (RRID:SCR_017016, v2.8.3) resulted in a highly fragmented genome of 993 Mb with an N50 of 459 Kb, while Hifiasm (RRID:SCR_021069, v0.15) produced a genome of 827 Mb comprising 779 contigs, but with an N50 of 46.1 Mb and an L75 of 14.

## Results for other *Macadamia* species

*M. jansenii* is an endangered species. It is one of four species in the *Macadamia* genus. Sequences of all four species were obtained using the same HiFi techniques and all gave similar, high-quality outcomes when assembled (Table 2).

## Results for other plant species

Methods for sequencing plant genomes must be applied to genomes of various sizes and complexities. Macadamia is a basal eudicot. Other flowering plant genomes were sequenced to determine how widely applicable the results of this study would be in plant genome sequencing. Jojoba (*Simmondsia chinensis*), a core eudicot from the Caryopyllales, and avocado, a magnoliid, were compared. The three diverse genomes were all similar in size (700–1000 Mb). Many important plant genomes are in, or near this size range [14]. *M. jansenii* is an endangered species with relatively low heterozygosity, avocado has much greater heterozygosity [15], and jojoba has been reported to be a tetraploid [16]. Heterozygosity and polyploidy both complicate assembly [17, 18]. The quality of the assemblies was more contiguous (fewer contigs required to cover the genome) or similar (avocado) with less data in each of these cases when HiFi reads were used instead of the earlier continuous long reads (Table 4). The macadamia and jojoba genomes gave larger N50 values when using the HiFi (CCS) reads than with long reads (CLR). However, the N50 for the slightly larger genome of avocado was greater when using the long reads than when using that obtained with the HiFi reads. This suggest that the larger genome may have longer repeat regions that limit contig assembly in some parts of the genome with HiFi reads.



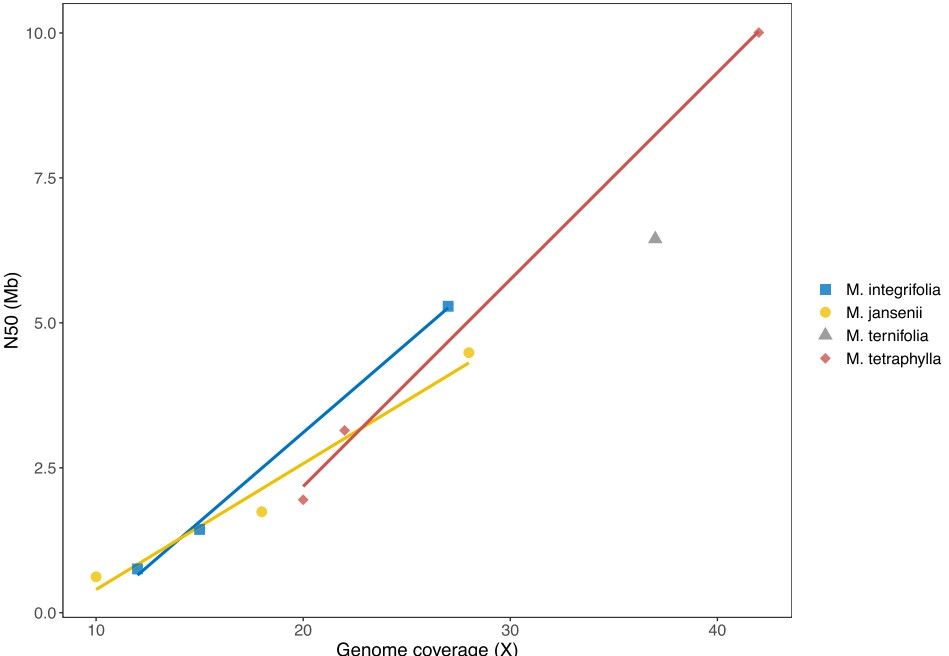

**Figure 2.** Influence of data volume on assembly for Macadamia species. N50 of contigs is plotted against the genome coverage. Genome sizes used to calculate coverage were; *M. integrifolia,* 895 Mb [20]; *M. janseni,* 780 Mb [1]; *M. tetraphylla,* 758 Mb [27] and *M. ternifolia,* 758 Mb (not known but assumed to be the same as *M. tetraphylla* owing to similar assembly size).

**Table 4.** Long read versus HiFi sequencing of other diverse species.

| | **Long reads** | | **HiFi** | |
| --- | --- | --- | --- | --- |
| **Parameter** | **Jojoba** | **Avocado** | **Jojoba** | **Avocado** |
| Total data (Gb) | 152 | 159 | 41.4 | 44 |
| Contig N50 (Mb) | 4.73 | 6.7 | 4.89 | 4.3 |
| Assembly length (Mb) | 1260 | 787 | 780 | 749 |
| Number of contigs (n) | 762 | 308 | 284 | 298 |
| BUSCO (%) | 99 | 99 | 98 | 98 |

## Impact of sequencing coverage on the assemblies

The length of the contigs assembled (Figure 2) was directly related to the volume of sequence data used. Analysis of four related *Macadamia* species gave a similar linear relationship between data volume and contig N50 for input of 10–40× genome coverage. The size of the contigs assembled showed a similar dependence on the amount of sequence data (genome coverage) across species, with the slope of the relationship varying for different species (Figure 3). The *Macadamia* genomes could be assembled with lower coverage. This may be a function of genome size, with their smaller genomes requiring less coverage to achieve a given N50. The larger genomes may contain a higher proportion of repetitive sequences that are difficult to assemble.

Assemblies based upon more data were slightly shorter in total length (Figure 4). This reduction was probably associated with removal of duplicated end sequences as contigs were joined. The high quality of these assemblies was confirmed by BUSCO values of more

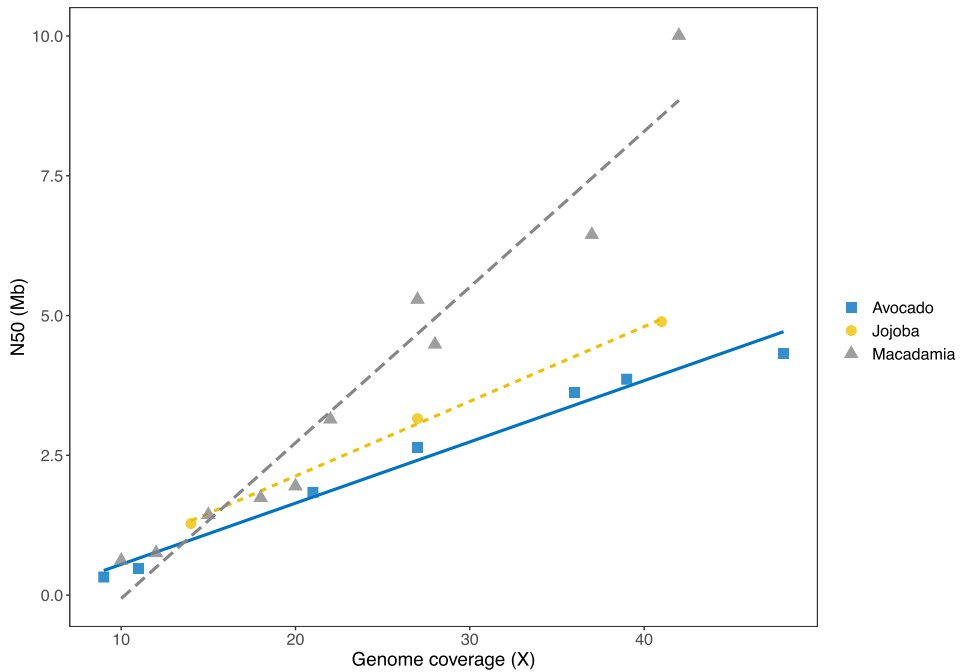

**Figure 3.** Influence of data volume on assembly for diverse species. N50 of contigs is plotted against the genome coverage. Genome sizes used to calculate coverage were jojoba 1003 Mb [22]; avocado 920 Mb [28], and as in Figure 2 for *Macadamia* species.

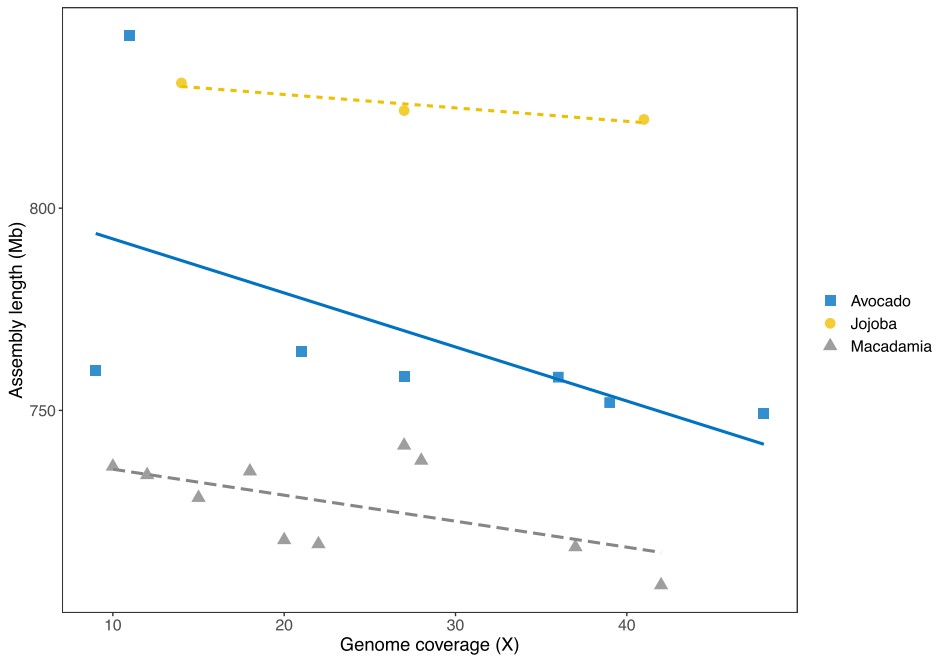

**Figure 4.** Decrease in length of total assembly as more genome coverage is used in the assembly.

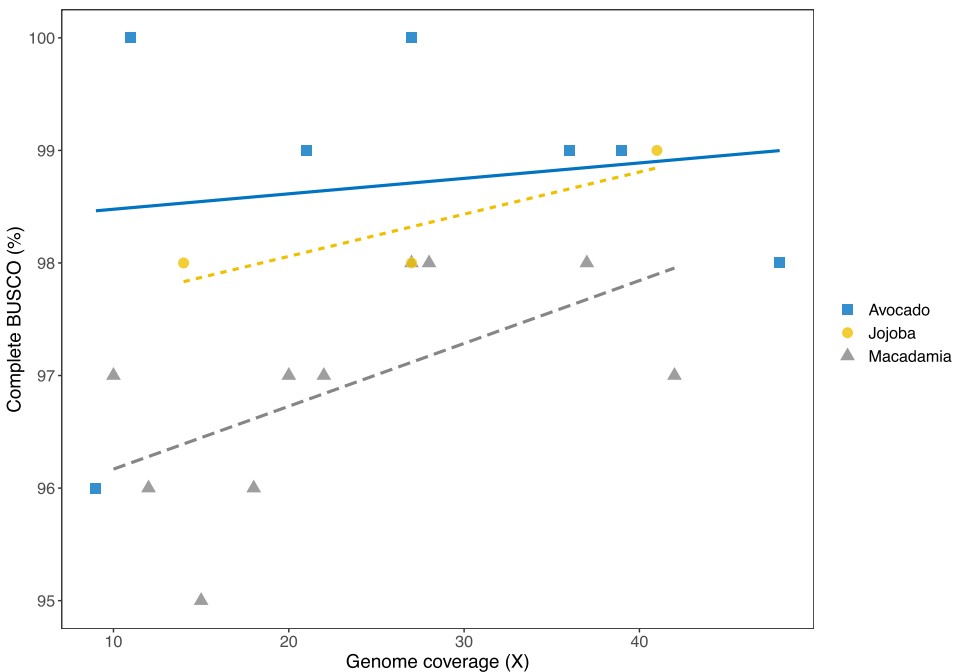

**Figure 5.** Improvement in genome completeness (BUSCO%) with genome coverage.

than 95%. Genome completeness was high in all cases but increased slightly when more data was used in the assembly (Figure 5).

These results were confirmed when applying these methods to sequencing the other phylogenetically diverse plant genomes with slightly larger genomes with greater genome complexity. In each case, N50 and completeness increased with data volume and genome size declined.

## Impact of the read length on the assemblies

The length of sequence reads was also expected to influence the assembly. Examination of size distribution of the six species showed that the length of the sequences varied slightly within the expected range – around 15 Kb for HiFi data. The minor differences in mean read length and numbers of longer reads did not explain the differences in the size of the contigs assembled (Supplementary Figure S1). This suggests that the different amounts of sequence data required to drive assembly to a particular level may be associated more with the complexity of the sequence. The close relationship between sequence volume and N50 for the four *Macadamia* species may reflect the similar sequence complexity of the species in this group. The jojoba and avocado genomes required more sequence data to reach the same level of assembly. The slightly larger genome size of these two species may be sufficient to explain this difference, owing to the likely higher proportion of repetitive sequence in the somewhat larger genomes.

## OXFORD NANOPORE TECHNOLOGIES UPDATES

ONT regularly releases updated basecalling software to convert raw electrical signal into sequence data. We repeated the basecalling of the ONT raw data of *M. jansenii* using

**Table 5.** ONT genome assembly statistics of *M. jansenii* using the Flye assembler, the pass reads and different Guppy basecaller versions.

| | Basecaller Assembler | | | |
| | Guppy v2.3.7 | Guppy v3.0.3 | | Guppy v4.0.11 |
| Parameter | Flye v2.5 | Flye v2.4.2 | Flye v2.5 | Flye v2.5 |
|---|---|---|---|---|
| Number of reads | 1597,353 | 1592,919 | | 1594,802 |
| Contig N50 (Mb) | 1.44 | 0.94 | 1.51 | 1.79 |
| Assembly length (Mb) | 817 | 845 | 811 | 798 |
| Number of contigs | 2996 | 4242 | 2855 | 2841 |
| Number of contigs (>10 kb) | 2300 | 3275 | 2088 | 1913 |
| BUSCO complete (%) | 66.8 | 51.4 | 75.1 | 79.1 |

different versions of the Guppy basecaller released in March 2019 (v2.3.7), April 2019 (v3.0.3) and June 2020 (v4.0.11). The assembly quality improved, as shown, by an increase in the assembly contiguity and in the number of complete BUSCOs before any polishing (Table 5). The assembly size decreased from 817 Mb to 798 Mb. Two versions of the Flye assembler were applied to the same basecalled sequence data set, which resulted in a marked increase in genome contiguity and completeness, as well as a reduced genome assembly size.

## REUSE POTENTIAL

These assemblies represent considerable advances over the highly fragmented genomes previously reported for these species [19–22]. Advances in long read sequencing using different platforms provide improving options for plant genome sequencing and assembly [23]. A recent comparison of these methods applied to rice genome sequencing showed strengths and weaknesses of both, with greater sequence accuracy in the Pac Bio assemblies and more contiguity in the ONT assemblies [4]. The resulting genome sequences can be evaluated for the best combination of sequence and assembly accuracy [24]. The results presented here show that contig size can be increased by adding more sequence reads to achieve a linear increase in N50. These extra data will result in slightly shorter total assembly lengths and improved completeness of the genomes. When combined with higher level assembly tools [25], the improved methods will support routine, rapid and efficient generation of highly accurate chromosome-level genome sequences of plant species [26].

## DATA AVAILABILITY

Sequence data from PacBio (Sequel) (RRID:SCR_017989), ONT (PromethION) (RRID:SCR_017987) and BGI (single-tube Long Fragment Read) (RRID:SCR_011114) analysis of *M. jansenii* was described by Murigneux *et al.* [1]. BGI, PacBio, ONT, and Illumina sequencing data generated in that study were deposited in the NCBI Sequence Read Archive (SRA) under BioProject PRJNA609013 and BioSample SAMN14217788. Accession numbers are as follows: BGI (SRR11191908), PacBio (SRR11191909), ONT PromethION (SRR11191910), ONT MinION (SRR11191911), and Illumina (SRR11191912). Assemblies and other supporting data are available from the *GigaScience* GigaDB repository [29]. Pac Bio HiFi reads described in this paper are deposited as CCS reads under NCBI BioProject ID Macadamia: PRJNA694456; Avocado: PRJNA694184 and Jojoba: PRJNA694450. Other data, further supporting this updated work are openly available in the GigaScience repository, GigaDB [30].

## DECLARATIONS
## LIST OF ABBREVIATIONS

BGI: Beijing Genomics Institute; bp: base pair(s); BUSCO: Benchmarking Universal Single-Copy Orthologs; CCS: circular consensus sequencing; CLR: continuous long reads; Gb: gigabase pair(s); Kb: kilobase pair(s); Mb: megabase pair(s); ONT: Oxford Nanopore Technology; PacBio: Pacific Biosciences.

## ETHICAL APPROVAL

Not applicable.

## CONSENT FOR PUBLICATION

Not applicable.

## COMPETING INTERESTS

The authors declare that they have no competing interests.

## FUNDING

This research received funding from the Hort Frontiers Advanced Production Systems Fund, Hort Frontiers Strategic Partnership, Hort Innovation, with University of Queensland and the Australian Government as part of National Tree Genomics Program, AS17000 Genomics Resources Toolbox, to R Henry; and from King Faisal University, Jojoba Genomics Project to RJ Henry, A Furtado and A Kharabian Masouleh.

## AUTHORS' CONTRIBUTIONS

Conceptualization: RJH, AF, VM, AM, IA; data curation: PS, OA, BA, ON, VM, AF; formal analysis: PS, OA, BA, ON, GM, VM, AM, AF; funding acquisition: RH, AF, IA, AM; investigation: PS, OA, BA, ON, NM, VM, AM, AF, RH; resources: IA, BT; supervision: IA, BT, NM, RH, AM, AF; writing of the original draft: RH, PS, ON, AM; writing, review and editing: all authors.

## ACKNOWLEDGEMENTS

The project was supported by the University of Queensland Research Computing Centre (RCC) and the University of Queensland Genome Innovation Hub.

## SUPPLEMENTARY DATA

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
