## [Reviewer Report]

Comments on revised manuscriptIn this revision, the authors answered some of my comments. My primary concern from the first version of the manuscript is whether the authors use the best possible de novo assemblers. They compare different technologies, and their conclusion relies on achieved results. 
In their previous paper, they showed that Canu consistently produces a much longer final sequence than other solutions. From the manuscript, it is not clear if they use methods for the removal of haplotypic duplications (ie. purge duplication). I deem that they should test at least one another assembler (ie. Flye) for error-prone reads. Flye is more resilient to haplotypic duplications.
Similarly, IPA is rarely used for hifi reads, and most of the authors use hifiasm. Even PacBio uses hifiasm in their analysis. 
The newest version of Flye and hifiasm are fast assemblers, so I deem their usage will not require significant computational resources. From above, I argue that the authors need to provide more results to support their claims.

---

## [Reviewer Report]

Comments on revised manuscriptFrom five of my comments, the authors answered only one (minor one). I cannot support publication of the manuscript in this form.

---

## [Reviewer Report]

Reviewer name and names of any other individual's who aided in reviewer Mile SikicDo you understand and agree to our policy of having open and named reviews, and having your review included with the published papers. (If no, please inform the editor that you cannot review this manuscript.)YesIs the language of sufficient quality?YesPlease add additional comments on language quality to clarify if needed
Are all data available and do they match the descriptions in the paper? YesAdditional CommentsAre the data and metadata consistent with relevant minimum information or reporting standards? See GigaDB checklists for examples <a href="http://gigadb.org/site/guide" target="_blank">http://gigadb.org/site/guide</a>YesAdditional CommentsIs the data acquisition clear, complete and methodologically sound?YesAdditional CommentsIs there sufficient detail in the methods and data-processing steps to allow reproduction?YesAdditional CommentsIs there sufficient data validation and statistical analyses of data quality? YesAdditional CommentsIs the validation suitable for this type of data?YesAdditional CommentsIs there sufficient information for others to reuse this dataset or integrate it with other data?YesAdditional CommentsIn their update to the previous study on the comparison of long read technologies for sequencing and assembly of plant genomes, Sharma et al. presented a follow-up analysis using a newer generation of basecallers for nanopore reads and PacBio HiFi reads. 
I argue that this study is an important update, but it is not suitable for publication in the current form. 
My major comments are the following:
1. It is not clear which version of the basecaller the authors used in assemblies related to Table 1 and Table 3
2. For phased assemblies, it is important to provide information about the size of alternative contigs
3. In Table 1, it would be great to have results for methods that do not phase assembly (i.e. Flye)
4. There is no explanation why authors use IPA instead of other HiFi assemblers, i.e. hifiasm, which from my experience, perform better than IPA
5. A sentence related to Table 3, “The quality of the assemblies was more contiguous with less data in each of these cases when HiFi reads were used instead of the earlier continuous long reads (Table 3).” is not clear. Following Table 3, assemblies achieved using long reads have similar or longer N50 and higher BUSCO score. Also, it is not clear which assembler was used for long reads
Any Additional Overall Comments to the AuthorRecommendationMajor Revision

---

## [Reviewer Report]

Reviewer name and names of any other individual's who aided in reviewer Chao BianDo you understand and agree to our policy of having open and named reviews, and having your review included with the published papers. (If no, please inform the editor that you cannot review this manuscript.)YesIs the language of sufficient quality?NoPlease add additional comments on language quality to clarify if needed
Are all data available and do they match the descriptions in the paper? YesAdditional CommentsAre the data and metadata consistent with relevant minimum information or reporting standards? See GigaDB checklists for examples <a href="http://gigadb.org/site/guide" target="_blank">http://gigadb.org/site/guide</a>YesAdditional CommentsIs the data acquisition clear, complete and methodologically sound?YesAdditional CommentsIs there sufficient detail in the methods and data-processing steps to allow reproduction?YesAdditional CommentsIs there sufficient data validation and statistical analyses of data quality? YesAdditional CommentsIs the validation suitable for this type of data?YesAdditional CommentsIs there sufficient information for others to reuse this dataset or integrate it with other data?YesAdditional CommentsAny Additional Overall Comments to the AuthorRecommendationMajor Revision